

# Eight single nucleotide polymorphisms and their association with food habit domestication traits and growth traits in largemouth bass fry (*Micropterus salmoides*) based on PCR-RFLP method

Jiao Cui[1], Zhou Jiang[1], Zerui Wang[1], Jiaqi Shao[1,2], Chuanju Dong[1], Lei Wang[1,3], Xuejun Li[1,3], Jinxing Du[2], Shengjie Li[1,2], Zhigang Qiao[1,3] and Meng Zhang[1,3]

[1] College of Fisheries, Henan Normal University, Xinxiang, China
[2] Pearl River Fisheries Research Institute, Chinese Academy of Fisheries Sciences, Key Laboratory of Tropical and Subtropical Fishery Resource Application and Cultivation, China Ministry of Agriculture, Guangzhou, China
[3] Engineering Technology Research Center of Henan Province for Aquatic Animal Cultivation, Xinxiang, China

## ABSTRACT

**Background:** The largemouth bass (*Micropterus salmoides*), an economically important freshwater fish species widely farmed in China, is traditionally cultured using a diet of forage fish. However, given the global decline in forage fish fisheries and increasing rates of waterbody pollution and disease outbreaks during traditional culturing, there is a growing trend of replacing forage fish with formulated feed in the largemouth bass breeding industry. The specific molecular mechanisms associated with such dietary transition in this fish are, nevertheless, poorly understood.

**Methods:** To identify single nucleotide polymorphisms (SNPs) related to food habit domestication traits and growth traits in largemouth bass fry, we initially genotyped fry using eight candidate SNPs based on polymerase chain reaction-restriction fragment length polymorphism (PCR-RFLP) method, with genetic parameters being determined using Popgen32 and Cervus 3.0. Subsequently, we assessed the associations between food habit domestication traits of largemouth bass fry and these SNPs using the Chi-square test or Fisher's exact test. Furthermore, we used a general linear model to assess the relationships between the growth traits of largemouth bass fry and these SNPs. The Pearson correlation coefficient between growth traits and the SNPs was also determined using bivariate correlation analysis in IBM SPSS Statistics 22. Finally, the phenotypic variation explained (PVE) by the SNPs was calculated by regression analysis in Microsoft Excel.

**Results:** The genotyping results obtained based on PCR-RFLP analysis were consistent with those of direct sequencing. Five SNPs (SNP01, SNP02, SNP04, SNP05, and SNP06) were found to be significantly correlated with the food habit domestication traits of fry ($P < 0.05$); SNP01 ($P = 0.0011$) and SNP04 ($P = 0.0055$) particularly, had showed highly significant associations. With respect to growth traits, we detected significant correlations with the two SNPs (SNP01 and SNP07)

Corresponding author
Meng Zhang,
zhangmeng@htu.edu.cn

($P < 0.05$), with SNP01 being significantly correlated with body length, and height ($P < 0.05$), and SNP07 being significantly correlated with body height only ($P < 0.05$).

**Conclusions:** Our findings indicated that the PCR-RFLP can be used as a low-cost genotyping method to identify SNPs related to food habit domestication and growth traits in largemouth bass, and that these trait-related SNPs might provide a molecular basis for the future breeding of new varieties of largemouth bass.

# INTRODUCTION

Food habit domestication is one of the important aspects of fishery cultivation. For carnivorous fish, switching to formulated feed can effectively reduce the pollution of the aquatic environment and the occurrence of diseases and also contributes to the conservation of marine resources (*Shao et al., 2022*). It is well established that during the fry stage, many fish species require live bait and do not readily switch to formulated feed. However, given that the use of live bait leads to high cultivation costs and may cause water pollution; the comprehensive replacement of forage fish with formulated feed is a foreseeable trend (*Welch et al., 2010*). Although studies on feed domestication have been carried out in mammals to analyze the related mechanism associated with feed transformation, there have been relatively few studies that have examined the molecular regulation mechanisms underlying the regulation of feed domestication in fish (*Wiener & Wilkinson, 2011*; *Zhao et al., 2010*). Among those studies that have been conducted, some have reported the possibility of "imprinting" fish with alternative components or nutritional levels in early life to improve their utilization in later life, although the specific regulatory mechanisms remain unclear (*Kwasek et al., 2021*; *Sammons, 2012*). In addition, based on the molecular mechanisms of feeding, some studies have attempted to identify the important regulatory factors associated with the consumption of formulated feed, to promote the comprehensive substitution of forage fish with formulated feed.

In this regard, the findings of previous studies have indicated that the feeding habits of carnivorous fish are influenced by pathways associated with the regulation of retinal photosensitivity, circadian rhythm, appetite control, learning, and memory (*He et al., 2013*). Given that genetic factors play an important role in switching to formulated feed, some studies have used candidate genes, such as *gh* (*growth hormone*) (*Dou et al., 2020*), *LPL* (*lipoprotein lipase*) (*Ma et al., 2018a*; *Yang et al., 2011*), *ghrelin* (*Liu et al., 2016*) and *PEP* (*pepsinogen*) (*Fang et al., 2011*), and association analysis to identify key molecular markers related to the traits involved in switching to formulated feed in carnivorous fish, with the aim of providing references for molecular assisted breeding of such traits.

The largemouth bass (*Micropterus salmoides*) is a typical predator fish native to North America, that has become an economically important freshwater fish in China (*Ma et al., 2020*). With gradual progress in the genetic improvement of largemouth bass,

important breakthroughs have been made with respect to the breeding for switching to formulated feed. For example, the growth and feed conversion traits of the new variety "Youlu No.3" have been significantly improved compared with those of its predecessor "Youlu No.1" and have made a significant contribution to the rapid development of the largemouth bass breeding industry (*Li et al., 2018*). However, the "Youlu No.3" fry must experience the succession of "live bait-dead bait-formulated feed". Thus, switching to formulated feed remains a relatively long process, and the success rate of switching to formulated feed still requires further improvement (*Zhao et al., 2019*). In addition, during the process of feeding habit domestication, the time of domestication varies markedly among individuals, and there are still individuals that fail to undergo successful domestication. Given that domestication has great influence on the survival rate and benefit of breeding, it is desirable to identify candidate markers related to switching to formulated feed in largemouth bass to improve the success rate of cultivation and shorten the period of switching to formulated feed.

Previously, we observed that some "Youlu No.3" do not need to transition *via* the "dead bait" stage prior to being able to consume formulated feed, and can fill the stomach. Accordingly, in the present study, we used "Youlu No.3" fry as the experimental material, and the process of switching to formulated feed was directly from "live bait-formulated feed" without "dead bait" transition stage. According to the degree of difficulty in receiving artificial formulated feed, we identified two extreme groups designated domesticated and non-domesticated. Subsequently, potential single nucleotide polymorphisms (SNPs) associated with the food habit domestication traits and growth traits were excavated by genotyping by sequencing (GBS). Among the SNPs identified, eight were successfully used to genotype fry based on polymerase chain reaction-restriction fragment length polymorphism (PCR-RFLP) for identification in large sample sizes, and we used these selected SNPs for further analysis of their association with feeding traits and growth traits. In the present study, we demonstrate that the largemouth bass fry can be directly transferred from the "live bait-the formulated feed" without the "dead bait" stage. The purpose of this study was to identify the SNPs related to the food habit domestication traits and growth traits of largemouth bass, and to screen out the largemouth bass that do not experience the "dead bait" stage and directly feed on formulated feed. Thus, our work provides a valuable reference for simplifying switching process of largemouth bass, and provides a theoretical basis for the further genetic improvement to increase the tolerance of largemouth bass to the formulated feed, which will promote the sustainable and healthy development of the largemouth bass breeding industry.

## MATERIALS AND METHODS

### Sample collection

The study was conducted according to the guidelines of the Declaration of Helsinki and approved by the Academic Committee of Henan Normal University (HNSD-2021-08-06). The "Youlu No.3" largemouth bass fry used in this study, which were the offspring of the random mating and natural spawning of 93 parental fish (51 females and 42 males), were obtained in May 2021 from a population cultured at the Lantian Aquaculture Professional

Cooperative (Zhoukou, China). Approximately 150,000 fertilized eggs were hatched on May 26 in a round tank (diameter = 1.5 m) fitted with a circulating water system (temperature = 25 ± 0.5 °C, DO = 8–9 mg/L). The fry were fed with artificially hatched brine shrimps (*Artemia salina*) from May 29 to June 20, during which time, the fry were gradually divided into 10 similar round tanks, corresponding with a reduction in feeding frequency decreased from 6 to 4 times a day.

Approximately 1,200 fish (20.34 ± 1.78 mm) were randomly selected from the 10 round tanks mentioned above and transferred into another round tank (the same as above) for 24 h of starvation. The powdered formulated feed with which the fry were subsequently provisioned was mixed with water and the bass were fed continuously for 2 h on June 21. The fry were subsequently anesthetized with MS-222 (3-aminobenzoic acid ethyl ester methanesulfonate; Sigma, Saint Louis, MO, USA), and growth data (body standard length from the front of the mouth to the base of the caudal fin and height) were measured using ImageJ software, expressed as the average of three consecutive readings (*Mishra et al., 2021*). Subsequently, the stomach of the fry were removed under a stereomicroscope and weighed. The fry were accordingly defined as non-domesticated (stomach/body weight < 8%, totaling 236 individuals) and domesticated (stomach/body weight > 24%, totaling 113 individuals) (*Zhao et al., 2019*). Every 96 juveniles were randomly selected from both the domesticated and non-domesticated groups, and their caudal fins were cut and preserved in absolute ethanol. Genomic DNA was then extracted using the Animal Genome Rapid Extraction Kit (Sangon, Shanghai, China). The quality and concentration of DNA were detected using 1% agarose gel electrophoresis and the NanoDrop 2000 (Thermo Fisher Scientific, Waltham, MA, USA). The DNA was dissolved in sterile water at 20 ng/μL, and stored at −20 °C.

## High-throughput sequencing and primer design

A total of 30 fry (15 individuals from each group) were randomly selected for the construction of GBS libraries and sequencing. Genomic DNA was digested with the restriction enzyme *Mse*I-*Nla*III-*Msp*I. PCR amplification was performed after adding adapters, and 215–240 base pair (bp) fragments were recovered to construct the GBS library. After the GBS library preparation, Illumina HiSeq PE150 sequencing was subsequently performed (Novogene, Beijing, China). The raw sequencing reads (BioProject: PRJNA769836) were filtered using fastp (v0.20.1) software with default parameters (*Chen et al., 2018*), and the resulting clean reads were aligned to the *Micropterus salmoides* reference genome (GenBank assembly accession: GCA_019677235.1) using BWA (v0.7.17) software (*Li & Durbin, 2010*). SNPs were called using the Genome Analysis Toolkit (GATK 4.1.9.0) software (*McKenna et al., 2010*). A total of 728 SNPs were found to differ significantly between the two groups ($P < 0.05$) based on preliminary analysis using a Chi-square test, of which 41 could be digested by *Eco*RV (3), *Hinf*I (20), *Pst*I (7), *Pvu*II (6), *Xba*I (3) and *Xho*I (2). On the basis of the flanking sequences and the results of restriction site analysis using an online website (http://www.detaibio.com/sms2/rest_summary.html), we designed 17 primer pairs using Primer Premier 5 software (*Lalitha, 2000*), however, only eight SNPs were successfully genotyped using PCR-RFLP (Table 1).

**Table 1 Primer information of eight SNPs in juvenile *M. salmoides*.**

| Locus | Flanking sequencing | Restriction enzyme | Primer information | | | |
|---|---|---|---|---|---|---|
| | | | Name | Sequence (5′-3′) | | Ta (°C) |
| SNP01 | CCCCT[G/A]CAGTC | *Pst*I | F1 | CAGTGGGAATGGTTATACATG | | 60 |
| | | | R1 | CTGAGAATGTAGCAGTAAAGTCC | | |
| SNP02 | CCAGC[T/A]GAGGA | *Pvu*II | F2 | CAACTAAAGGAAAGGCTGATT | | 54 |
| | | | R2 | GGAAGACGAGAAACACGAAA | | |
| SNP03 | GACTC[G/A]AGATC | *Xho*I | F3 | TCGGGAAATCCGTGTTGA | | 58 |
| | | | R3 | CAGGTTAAACTTTGTTCCTGGTC | | |
| SNP04 | AGATT[T/C]GTCAC | *Hinf*I | F4 | GCCCAAGAGGACACTTAGAT | | 56 |
| | | | R4 | TTTTTACGAGATAGGGTAGACAT | | |
| SNP05 | CAGAG[G/C]AGCTG | *Pvu*II | F5 | CTTCATGCAGTTGGGTATT | | 48 |
| | | | R5 | AATACTTATGTTTGCCCTTG | | |
| SNP06 | AGTGA[C/T]ATCAC | *Eco*RV | F6 | TTGTGTTATTTGAAGAGTAACTTAAC | | 62 |
| | | | R6 | AATCTACCCATCTACAGTCCC | | |
| SNP07 | CATAG[C/A]TGCAG | *Pst*I | F7 | ATCCTACAATTGGATGTAACTT | | 60 |
| | | | R7 | AAAGCCCAACTATTACCC | | |
| SNP08 | CAGCT[C/A]TAGAG | *Xba*I | F8 | ATCTCCCACGCCAAGTCA | | 63 |
| | | | R8 | AAAATCCAAGTGCGGTCTG | | |

## PCR-RFLP genotyping and identification

The PCR-RFLP method was used for SNPs genotyping. PCR was performed in a 20 μL volume mixture containing 0.5 μmol/L primer, 10 μL of 2 × Master Mix (Vazyme, Nanjing, China), 60 ng template DNA, and deionized water added to 20 μL. The PCR conditions were as follows: pre-denaturation at 95 °C for 3 min; denaturation at 95 °C for 30 s, annealing time for 30 s (the annealing temperature (Ta) is shown in Table 1), extension at 72 °C for 45 s for a total of 34 cycles, and a final extension at 72 °C for 5 min. The PCR products were detected using 1% agarose gel electrophoresis, and the qualified PCR products were digested. The restriction enzymes (Sangon, Shanghai, China) corresponding to the eight SNPs are listed in Table 1. The enzyme restriction system was performed in a 10 μL volume containing 5 μL of PCR product, 0.5 μL restriction enzyme, 1 μL of 10 × Speedy One Buffer, and deionized water supplemented to 10 μL. The reaction was then performed at 37 °C for 45 min. The fragment size of the digested product was detected using 2% agarose gel electrophoresis. PCR products corresponding to the different genotypes of each SNP were randomly selected for direct sequencing.

## Statistical analysis

Microsoft Excel (Microsoft Corp., Redmond, WA, USA) was used for statistical analysis of the morphological data and genotyping results. Analyses of the observed heterozygosity (Ho), expected heterozygosity (He), and the polymorphic information content (PIC) were performed using Cervus 3.0 software (*Botstein et al., 1980*; *Kalinowski, Taper & Marshall, 2007*). Popgen32 software was used to analyze the Hardy-Weinberg equilibrium (*Yeh &*
*Boyle, 1996*). The correlation between genotypes at each locus and food habit domestication traits of the fry was analyzed using the Chi-square or Fisher exact test in R software. The general linear model in IBM SPSS Statistics 22 (IBM Corp., Armonk, NY, USA) was used to analyze the correlation between the genotypes at each locus and body height, and length of largemouth bass fry (*Ma et al., 2018b*). The bivariate correlation analysis was used to analyze the correlation between the genotypes of each locus and growth traits in IBM SPSS Statistics 22 (*Weaver & Wuensch, 2013*), and the phenotypic variation explained (PVE) was calculated using regression analysis in Microsoft Excel.

## RESULTS

### Comparative analysis of direct sequencing peak and PCR-RFLP
The genotyping results revealed that all eight selected SNPs were successfully genotyped using the PCR-RFLP method. Each locus had a homozygous wild genotype, heterozygous mutant genotype, and homozygous mutant genotype. The PCR products of different genotypes at each locus were randomly selected and subjected to direct sequencing. Comparison of a sequencing peak map with the banding patterns of the PCR-RFLP products on agarose (Fig. 1), confirmed the direct sequencing results were consistent with the PCR-RFLP genotyping results, thereby indicating the applicability of the PCR-RFLP approach for SNP genotyping.

### Polymorphism analysis of eight SNPs
The genotype frequencies of eight SNPs were analyzed using Microsoft Excel. The results revealed that the Ho of eight SNPs ranged from 0.3490 to 0.5417, He ranged from 0.3514 to 0.5013, and PIC ranged from 0.2891 to 0.3750. All eight SNPs were moderately polymorphic ($0.25 \leq PIC < 0.5$), and SNP03 deviated significantly from the Hardy-Weinberg equilibrium (Table 2).

### Correlation analysis between eight SNPs and food habit domestication traits
The Chi-square test or Fisher's exact test were used to analyze the correlation between the eight SNPs and the food habit domestication traits of the fish fry (Table 3). The results showed that three SNPs (SNP02, SNP05, and SNP06) were significantly correlated with food habit domestication traits ($P < 0.05$), with PVE values of 3.29, 0.02 and 3.11, respectively. In addition, two SNPs (SNP01 and SNP04) were highly significantly correlated with food habit domestication traits ($P < 0.01$), with PVE values of 7.08 and 5.39, respectively.

### Associations between eight SNPs and growth traits
The analysis of the correlation between eight SNPs and growth traits of the fry using the general linear model revealed that two SNPs (SNP01 and SNP07) were significantly associated with the growth traits of largemouth bass fry (Table 4). The body height differed significantly with respect to the three genotypes of SNP01 ($P < 0.05$), the Pearson correlation coefficient between SNP01 and body height was 0.291 and PVE value was 8.45.

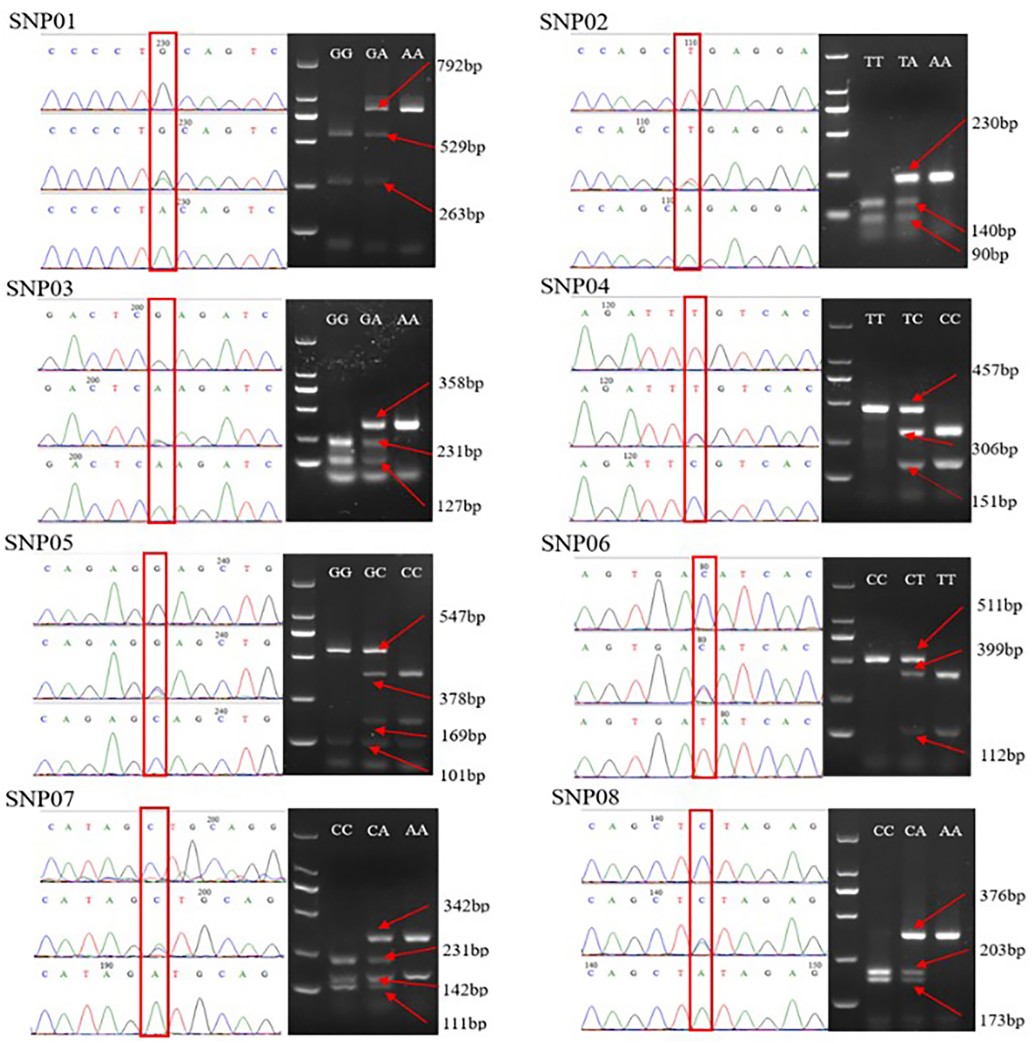

**Figure 1 Comparison of the sequencing peaks of different genotypes of SNP01-SNP08 and PCR-RFLP.** The red box indicates the mutation site, and different sizes of enzyme fragments are on the right. The GG of SNP05 and the AA of SNP07 have two bands after digestion due to the existence of enzyme sites in the selected flank sequence, which is independent of specific SNP.

The body length showed a significant difference only between the GG and GA genotypes ($P < 0.05$), the Pearson correlation coefficient between SNP01 and body length was 0.172 and PVE value was 2.95. Additionally, the body height of the SNP07 AA genotype had a significantly high correlation with that of the CC genotype ($P < 0.05$). The Pearson correlation coefficient between SNP07 and body height was 0.156, and PVE value was 2.43.

## DISCUSSION

### Application of PCR-RFLP for SNPs genotyping of largemouth bass fry

As the third generation of molecular genetic markers, SNPs have broad application prospects in animal and plant breeding because of their large number, wide distribution, and considerable effects on phenotypes (*Lambert et al., 2016*; *Siccha-Ramirez et al., 2018*;

**Table 2 Genotype frequency and genetic parameters of eight SNPs in juvenile *M. salmoides*.**

| Locus | Genotype frequency | | | Ho | He | PIC | PHWE |
|---|---|---|---|---|---|---|---|
| SNP01 | GG(0.4896) | GA(0.4219) | AA(0.0885) | 0.4219 | 0.4207 | 0.3316 | 0.9684 |
| SNP02 | TT(0.0938) | TA(0.3490) | AA(0.5573) | 0.3490 | 0.3936 | 0.3155 | 0.1147 |
| SNP03 | GG(0.0990) | GA(0.5417) | AA(0.3594) | 0.5417 | 0.4673 | 0.3575 | 0.0270* |
| SNP04 | TT(0.3958) | TC(0.4948) | CC(0.1094) | 0.4948 | 0.4602 | 0.3536 | 0.2957 |
| SNP05 | GG(0.3021) | GC(0.5313) | CC(0.1667) | 0.5313 | 0.4921 | 0.3704 | 0.2692 |
| SNP06 | CC(0.3906) | CT(0.4740) | TT(0.1354) | 0.4740 | 0.4687 | 0.3582 | 0.8750 |
| SNP07 | CC(0.2344) | CA(0.5365) | AA(0.2292) | 0.5365 | 0.5013 | 0.3750 | 0.3298 |
| SNP08 | CC(0.5833) | CA(0.3802) | AA(0.0365) | 0.3802 | 0.3514 | 0.2891 | 0.2533 |

Note:
* Significant differences ($P < 0.05$).

*Zhang et al., 2019*). However, SNPs have their own application limitations, such as its relatively high genotyping costs. Fortunately, several methods have been developed and improved to decrease SNP genotyping cost, including allele-specific PCR (AS-PCR), SNaPshot, and PCR-RFLP (*Zhao et al., 2017*). The PCR-RFLP is a cost-effective method and has been successfully applied in SNP genotyping of many species (*Forche, Steinbach & Berman, 2009*; *Jiang et al., 2021*). In the 1980s, *Botstein et al. (1980)* used DNA RFLP to construct a genetic linkage map of human genes, which pioneered the use of DNA polymorphic genetic markers. However, the procedures associated with this technique are notably complex, and there are certain complications, such as the increase, decrease and movement of enzyme digestion sites, which limits the widespread application of RFLP markers. PCR-RFLP combines the advantages of PCR and RFLP, and this combined technique is frequently used as the method of choice in analyses of the genetic variation of genomic DNA to reveal SNPs loci or in the use of known SNPs loci for genotyping is increasingly favored. For example, *Viana et al. (2007)* designed a PCR-RFLP strategy for the G/A mutation site at base pair 1,440 bp of the human *CXCR2* gene and successfully genotyped this, whereas *Ma et al. (2011)* also used this technique to identify a SNP site on the polymorphism of the partial sequence of the antimicrobial peptide gene *SCY2* in *Scyllapar amamosain*, which was not found by direct sequencing, thereby confirming that PCR-RFLP has considerable applicability in detecting molecular genetic variations.

In terms of SNP genotyping, PCR-RFLP technology has clear advantages compared to direct sequencing, including low cost, rapidity, and reliable analytical results. Nevertheless, it has certain limitations. If SNPs cannot form restriction sites, PCR-RFLP cannot be used directly for genotyping. Even if SNPs can form restriction sites, the genotyping cost of each site may vary greatly due to the different costs of restriction enzymes (*Xu & Shen, 2003*). For example, in a 20 μL enzyme restriction system, the genotyping cost of SpeedyCut *EcoR*I was 0.5 CNY/site, while the genotyping cost of SpeedyCut *Fsp*I was six CNY/site. In addition, the PCR-RFLP method is susceptible to the type and number of restriction enzyme sites in flanking sequences (*Yan et al., 2022*). For example, in the present study, there was a G/C mutation at SNP05 in this study, which *Pvu*II could restrict, and the PCR product fragment containing this SNP site was 648 bp. However, another
**Table 3 Correlation analysis between eight SNPs and food habit domestication traits in *M. salmoides* fry.**

| Locus | Genotype | Number (Domesticated) | Number (Non-domesticated) | Chi-square/ Fisher | *P* value | Cramér's V correlation coefficient | PVE (%) |
|-------|----------|------------------------|----------------------------|---------------------|-----------|-------------------------------------|---------|
| SNP01 | GG | 59 | 35 | 13.670 | 0.001** | 0.267** | 7.08 |
|       | GA | 33 | 48 | | | | |
|       | AA | 4 | 13 | | | | |
| SNP02 | TT | 11 | 7 | 7.621 | 0.022* | 0.199* | 3.29 |
|       | TA | 41 | 26 | | | | |
|       | AA | 44 | 63 | | | | |
| SNP03 | GG | 14 | 5 | 4.740 | 0.093 | 0.157 | 1.00 |
|       | GA | 49 | 55 | | | | |
|       | AA | 33 | 36 | | | | |
| SNP04 | TT | 28 | 48 | 10.394 | 0.006** | 0.233** | 5.39 |
|       | TC | 53 | 42 | | | | |
|       | CC | 15 | 6 | | | | |
| SNP05 | GG | 34 | 24 | 6.901 | 0.032* | 0.190* | 0.02 |
|       | GC | 42 | 60 | | | | |
|       | CC | 20 | 12 | | | | |
| SNP06 | CC | 47 | 28 | 7.901 | 0.020* | 0.203* | 3.11 |
|       | CT | 38 | 53 | | | | |
|       | TT | 11 | 15 | | | | |
| SNP07 | CC | 19 | 26 | 4.604 | 0.100 | 0.155 | 2.11 |
|       | CA | 49 | 54 | | | | |
|       | AA | 28 | 16 | | | | |
| SNP08 | CC | 54 | 58 | 3.579 | 0.202 | 0.139 | 0.69 |
|       | CA | 36 | 37 | | | | |
|       | AA | 6 | 1 | | | | |

**Note:**
Asterisks (* and **) indicate significant (*P* < 0.05) and extremely significant differences (*P* < 0.01), respectively. "PVE" represents the phenotypic variation explained.

restriction enzyme site in the PCR product was recognized by *Pvu*II, which led to the 101 and 547 bp bands. Therefore, there were two bands of 101 and 547 bp in the digested products of the homozygous wild-type GG, and *Pvu*II could completely digest the 547 bp PCR product of the homozygous mutant CC to produce 169 and 378 bp, giving three bands of 101, 169, and 378 bp. In the PCR products of heterozygous mutant type GC, only part of the 547 bp could be digested by *Pvu*II to produce 169 and 378 bp, giving four bands of 101, 169, 378, and 547 bp. Furthermore, the PCR-RFLP method usually requires enzyme restriction after PCR amplification, which can easily cause pollution and affect the genotyping results due to the open operation. In addition, each locus must be digested after PCR amplification, which is more suitable for SNP genotyping with a small number of loci and a medium/large sample size.

**Table 4 Correlation analysis between the different genotypes of eight SNPs and growth traits in _M. salmoides_ fry.**

| Locus | Genotype | Number | Body height (mm) | Pearson correlation coefficient | PVE (%) | Body length (mm) | Pearson correlation coefficient | PVE (%) |
|---|---|---|---|---|---|---|---|---|
| SNP01 | GG | 94 | 5.647 ± 0.720[a] | 0.291** | 8.45 | 20.643 ± 1.936[a] | 0.172* | 2.95 |
| | GA | 81 | 5.393 ± 0.574[b] | | | 20.113 ± 1.604[b] | | |
| | AA | 17 | 4.971 ± 0.607[c] | | | 19.774 ± 1.478[ab] | | |
| SNP02 | TT | 18 | 5.423 ± 0.702 | −0.019 | 0.04 | 20.288 ± 1.831 | −0.074 | 0.55 |
| | TA | 67 | 5.534 ± 0.680 | | | 20.623 ± 1.709 | | |
| | AA | 107 | 5.456 ± 0.678 | | | 20.176 ± 1.816 | | |
| SNP03 | GG | 19 | 5.737 ± 0.621 | 0.071 | 0.51 | 20.697 ± 1.258 | 0.067 | 0.45 |
| | GA | 104 | 5.442 ± 0.650 | | | 20.351 ± 1.782 | | |
| | AA | 69 | 5.467 ± 0.730 | | | 20.232 ± 1.913 | | |
| SNP04 | TT | 76 | 5.397 ± 0.701 | 0.074 | 0.54 | 20.452 ± 1.731 | −0.082 | 0.67 |
| | TC | 95 | 5.548 ± 0.678 | | | 20.359 ± 1.849 | | |
| | CC | 21 | 5.475 ± 0.595 | | | 19.871 ± 1.675 | | |
| SNP05 | GG | 58 | 5.508 ± 0.678 | 0.025 | 0.06 | 20.464 ± 1.804 | 0.110 | 1.21 |
| | GC | 102 | 5.470 ± 0.718 | | | 20.451 ± 1.805 | | |
| | CC | 32 | 5.463 ± 0.561 | | | 19.776 ± 1.613 | | |
| SNP06 | CC | 75 | 5.472 ± 0.717 | 0.042 | 0.18 | 20.087 ± 1.843 | −0.074 | 0.55 |
| | CT | 91 | 5.535 ± 0.647 | | | 20.575 ± 1.736 | | |
| | TT | 26 | 5.312 ± 0.671 | | | 20.265 ± 1.733 | | |
| SNP07 | CC | 45 | 5.314 ± 0.622[b] | 0.156* | 2.43 | 20.141 ± 1.514 | 0.070 | 0.49 |
| | CA | 103 | 5.492 ± 0.699[ab] | | | 20.360 ± 1.833 | | |
| | AA | 44 | 5.621 ± 0.663[a] | | | 20.507 ± 1.936 | | |
| SNP08 | CC | 112 | 5.477 ± 0.692 | 0.027 | 0.07 | 20.277 ± 1.647 | 0.045 | 0.21 |
| | CA | 73 | 5.465 ± 0678 | | | 20.422 ± 1.995 | | |
| | AA | 7 | 5.703 ± 0.506 | | | 20.554 ± 1.759 | | |

Note:
Different superscript letters in a column of each locus indicate significant a difference ($P < 0.05$), * and ** indicate significant ($P < 0.05$) and extremely significant differences ($P < 0.01$), respectively. "PVE" represents the phenotypic variation explained.

## Association analysis of food habit domestication traits and growth traits in largemouth bass fry

The largemouth bass is an economically important freshwater fish in China, wherein it has been widely cultured in recent years. During the culturing process, the cost of rearing can be effectively reduced by directly transfer from "live bait" to "formulated feed" without transitioning through the "dead bait" stage. Although largemouth bass fry do not readily switch to formulated feed, while some studies have shown that an early transfer to formulated feed can increase food intake and improve the later growth performance (_Skudlarek, Coyle & Tidwell, 2013_). Therefore, improving the success rate and shortening the period of switching to formulated feed of largemouth bass would be advantageous (_Ehrlich et al., 1989_). In this regard, the use of molecular markers to screen largemouth bass that can be easily switched to formulated feed without "dead bait" stage can effectively solve the problems of breeding environment contamination and disease associated with "dead bait" stage in the cultivation process of largemouth bass. With the publication of the

largemouth bass genome and the decrease in high-throughput sequencing costs (*Sun et al., 2021*), it is possible to use high-throughput sequencing technology to identify molecular markers related to the economic traits of largemouth bass.

In this study "Youlu No.3", which did not experience the "dead bait" stage, was used as the experimental material to screen SNPs related to food habit domestication. We identified five SNPs related to food habit domestication traits and two SNPs related to growth traits were successfully verified. The results showed that there were significant differences in the body height traits among the three genotypes of SNP01 (GG > GA > AA, $P < 0.05$), with respect to body height, and between the two genotypes of SNP01 (GA > AA, $P < 0.05$), with respect to body length, whereas the two genotypes of SNP07 were found to be associated with body length, which may be related to growth rate differences at different stages (*Gong et al., 2022*; *Jiang et al., 2020*). Further research on the other potential food habit domestication-related SNPs based on GBS data is required to verify the potential application of marker-assisted selection for largemouth bass in the future.

## CONCLUSIONS

In summary, the PCR-RFLP method was successfully and accurately applied in the genotyping of eight randomly selected SNPs. Five food habit domestication-related SNPs and two growth-related SNPs were identified in largemouth bass fry. The results of the present study suggest that the PCR-RFLP might be a low-cost and effective method for excavating trait-related SNPs, especially using "small sample/big data" to excavate and then the correlation is verified by "slight amount markers/big sample". Overall, our findings provide candidate markers for further genetic improvement of the related traits of largemouth bass.

## ACKNOWLEDGEMENTS

Offspring and fry were kindly provided by the Engineering Technology Research Center of Henan Province for Aquatic Animal Cultivation.

### Funding

This work was supported by the National Natural Science Foundation of China (32102776), Ph.D. Foundation of Henan Normal University (qd19067), and the Central Public-interest Scientific Institution Basal Research Fund (CAFS-2021SJ-CG1). This work was also supported by the High Performance Computing Center of Henan Normal University. The funders had no role in study design, data collection and analysis, decision to publish, or preparation of the manuscript.

### Grant Disclosures

The following grant information was disclosed by the authors:
National Natural Science Foundation of China: 32102776.

 

Henan Normal University: qd19067.
Central Public-interest Scientific Institution Basal Research Fund: CAFS-2021SJ-CG1.

## Competing Interests

The authors declare that they have no competing interests.

## Author Contributions

- Jiao Cui performed the experiments, analyzed the data, prepared figures and/or tables, authored or reviewed drafts of the article, and approved the final draft.
- Zhou Jiang performed the experiments, analyzed the data, prepared figures and/or tables, authored or reviewed drafts of the article, and approved the final draft.
- Zerui Wang analyzed the data, prepared figures and/or tables, and approved the final draft.
- Jiaqi Shao analyzed the data, prepared figures and/or tables, and approved the final draft.
- Chuanju Dong conceived and designed the experiments, authored or reviewed drafts of the article, and approved the final draft.
- Lei Wang analyzed the data, authored or reviewed drafts of the article, and approved the final draft.
- Xuejun Li analyzed the data, authored or reviewed drafts of the article, and approved the final draft.
- Jinxing Du analyzed the data, authored or reviewed drafts of the article, and approved the final draft.
- Shengjie Li conceived and designed the experiments, authored or reviewed drafts of the article, and approved the final draft.
- Zhigang Qiao conceived and designed the experiments, authored or reviewed drafts of the article, and approved the final draft.
- Meng Zhang conceived and designed the experiments, analyzed the data, prepared figures and/or tables, authored or reviewed drafts of the article, and approved the final draft.

## Animal Ethics

The following information was supplied relating to ethical approvals (*i.e.*, approving body and any reference numbers):

The Academic Committee of Henan Normal University provided full approval for this research (HNSD-2021-08-06).

## Data Availability

The raw sequence read data is available at BioProject: PRJNA769836.

## Supplemental Information

Supplemental information for this article can be found online at http://dx.doi.org/10.7717/peerj.14588#supplemental-information.

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
