# Peer review of "Eight single nucleotide polymorphisms and their association with food habit domestication traits and growth traits in largemouth bass fry (Micropterus salmoides) based on PCR-RFLP method"

_PeerJ, doi:10.7717/peerj.14588_

## Round 0.1 · original submission · Major Revisions

The manuscript describes eight SNPs that are related to two essential traits (food habit domestication & growth) of juveniles largemouth bass. Although in general, the manuscript provide important baseline support to the breeding and aquaculture of this species, there are some major concerns raised by the reviewers that I think are valuable in the improvement of the quality of this manuscript. In particular, the correlation analysis (and its results) is questionable. Also, do improve on the Introduction section as the current Introduction lacks depth. Similarly, the Discussion section (especially 4.1) should be discussed together with the support of other studies. For example, line 203-221 has no citation.

Reviewer 1 ·

Basic reporting

• Please show correlation coefficient between genotype and food habit domestication in table 3 as well as the correlation between genotype and growths in table 5.
• Line 147, Please check if the reference is correct because the referred paper does not state the correlation analysis.

Experimental design

• Please give information regarding fish production, i.e., number of broodstock, the number of families produced, and the number of offspring. This is crucial in term of association test because the fish used in the experiment should be sampled from many families (representing the studied population) rather than a few families.
• Line 95-97, Please give the number of the fish. By “a new aquaculture system”, were they reared in the same tank/pond or with replicates?
• Line 98, By length, did you mean standard length or total length?
• Line 116, How many SNPs with distinctly different frequencies between groups did you find? Because authors sampled only eight SNPs from these SNPs, without considering these SNPs in the analysis might cause overestimate of SNP effect resulting in false positive.
• Line 144-147, In this study, authors use correlation between genotype and phenotype (food habit domestication and growths) as the criteria to test association of a SNP with the traits. However, the test relies on only phenotype information and ignores environmental factors, SNPs effect in other loci and population structure, and may lead to false positive. I suggest a more powerful statistical test which utilize mixed model should be applied in this study and compared with your result based on correlation. Also, it would be great if you could increase the number of SNPs used in this study i.e., use all SNPs with high frequency difference between groups.

Validity of the findings

• Line 263-270, Authors stated that “the finding of this study provide candidate markers for further genetic improvement”. Despite its significance, it might not contribute much to the traits. If authors would like to claim this context, authors should further examine their effect or the proportion of SNP (genetic) variance that contributes to phenotypic variance.

Additional comments

This manuscript aims to investigate SNPs that associate with adaptation to formulated feed in largemouth bass. Authors present interesting SNP-selection criteria by categorizing the fish in feed-adaptable and nonadaptable groups, then randomly selected SNPs based on difference frequency between groups. Authors also propose PCR-REML as an alternative, cost-saving method for genotyping targeted SNPs. However, some points need to be clarified before publication, as in my comments.

·

Basic reporting

In this study, eight SNPs from largemouth bass were detected for the relationship to food habit domestication and growth traits. Five SNPs were found significantly correlated with the food habit domestication trait and two SNPs were significantly correlated with the growth traits. This work provided a reference for molecular marker assisted breeding of largemouth bass. In the manuscript, English expression is clear, sufficient field background is provided.However, in the manuscript the methods are not expressed clearly, and some sentences are imprecise, and will make the readers confused.

Experimental design

The experimental design has some problems.
1)Why do you select the new variety “Youlu No.3” as experimental fish? “Youlu No.3” is a selected as a new variety of largemouth bass adapting to formulated diets.
2)In Line 101-102, “96 juveniles were randomly selected from domesticated and non- domesticated”. How many fish were measured all for the food habit domestication traits? And How many are domesticated, and how many are non- domesticated? Please add the data, and they are very important.
3)The correlation analysis between the different genotypes of eight SNPs and growth traits should be carried out among a fish population with significant difference of growth traits. All of the 192 individuals has approximate 20mm body length with SD of 0.176 to 0.664 mm. The population with same size fishes is not suitable for the correlation analysis. This part should be deleted.
4) On the other hand, growth traits were measured after they were fed 2h, some fish are extreme full, and some are empty. The experimental fish are clearly inappropriate to measure body weight.

Validity of the findings

For effects of body length and body height traits on body weight, suppose two fish with the same body weight, body length and height, one is in domesticated group (stomach is full), the other is in non- domesticated group (stomach is empty). Then minimum weight difference between the domesticated fish and the non- domesticated fish was 16% of body weight, the acurracy of results will be affected seriously.
All analysis about the body weight measured after feeding 2h should be deleted.

Additional comments

1)“Juvenile fish” does not mean the fish with 20mm-body-length. Please see the reference “James D. Bowker, Daniel Carty, Charlie E. Smith & Silas R. Bergen (2011): Chloramine-T Margin-of-Safety Estimates for Fry, Fingerling, and Juvenile Rainbow Trout, North American Journal of Aquaculture, 73:3, 259-269.”
2)In Table 5, the unit of body is mg or g?

·

Basic reporting

The manuscript is written in a a well organized manner with proper subtopics. The introduction clearly highlights the issue and problem as well as the literature review related to the scope of the research. However the objective is not clearly stated in the manuscript. It is suggested that the author could clearly state the objective of the study.

Experimental design

The experimental design covers the whole aspects of the research. The approach is relevant and significant which can lead to the attainable of the objective.
However, in understanding the molecular aspect of the issues highlighted by the author, we need to understand that the set up of the culture is very important and can affect the findings.
Due to this it is suggested that the author would include some information regarding the culture of the studied species in section 2.1 as follows :
1) How was the larvae produce (natural spawning or artificial spawning)
2) how many parents were used to produced the eggs. Is it from one pair or several pairs
3) It was stated in the manuscript that juveniles reaching 2 cm were collected and transferred. It would be better if the approximate duration of the culture to reach 2 cm was mentioned
4) What is the stocking density of the cultured larvae and also for the juveniles that was transferred
5) Is the new aquaculture system used to maintain 2 cm juveniles is the same culture system (circulating water system) as the one used for larvae rearing and what is the size of the culture system.
6) how was the fish killed before the stomach was removed? Is it using pithing techniques or some other technique

Validity of the findings

no comments

Additional comments

no comments

---

## Round 0.2 · Minor Revisions

I agree with the reviewers that the revised manuscript still has some room for improvement before ready for acceptance. Do kindly check all relevant data, including those in the supplementary files, as some discrepancies were pointed out by reviewer 1.

·

Basic reporting

In this study, eight SNPs from largemouth bass were detected for the relationship to food habit domestication and growth traits. Five SNPs were found significantly correlated with the food habit domestication trait and two SNPs were significantly correlated with the growth traits. This work provided a reference for molecular marker assisted breeding of largemouth bass. However, in the manuscript there are still some obvious mistakes that need to be corrected before acceptance.

Experimental design

The correlation analysis between the different genotypes of eight SNPs and growth traits should be carried out among a fish population with significant difference of growth traits. All of the 192 individuals has approximate 20mm body length are not very suitable. The molecular markers can only be used as a reference.

Validity of the findings

There is some obvious mistakes in the Supplemental files. For example, individual 1, Body Length(mm) is 23.670, Body Height(mm) is 0.15004, and Body Weight(mg) is 7.03. Please correct.

Additional comments

Please check all the manuscript carefully.

·

Basic reporting

The author has made the necessary amendments to the manuscript

Experimental design

The author has made the necessary amendments

Validity of the findings

no comment

Additional comments

Line 99 : there was a grammatical error.
It is suggested that the sentence should be :Accordingly, in this study, we used "Youlu No. 3" fry as the experimental material, ........

---

## Round 0.3 · accepted · Accept

Thank you for following through with the revision process. The manuscript now looks ready for publication.